# How Do Hospitality Workers Perceive Their Work Skills before and after the Lockdown Imposed by the COVID-19 Pandemic?

Carla Magalhães , Arthur Araújo * and Maria Isabel Andrés-Marques

Transdisciplinary Research Center on Innovation & Entrepreneurship Ecosystems (TRIEE), Faculty of Economic, Social and Business Sciences (FCESE), Lusofona University, 4000-098 Porto, Portugal
* Correspondence: p5706@mso365.ulp.pt

**Abstract:** The present research aims to compare hospitality workers' confidence about the perceived need to improve their soft and hard skills during and after the lockdown period. With this purpose, a questionnaire was applied to a sample of workers from four- and five-star hotels located in the historic centre of Porto (Portugal). Data collection took place in two different time periods: April 2020 and April 2022. The findings evidence that workers are more confident of their skills but are also more aware of the need to develop them, particularly their soft skills. This is likely related to the context of remote work, which intensified the need to learn, and evidenced the necessity of skills such as teamwork and adaptation or flexibility, which showed the greatest increase in perceived need to improve. Despite such an increase, the competencies workers feel like they need to improve the most are still hard, i.e., linguistic and digital. Accordingly, those in which workers are the most confident are soft, i.e., teamwork, interpersonal relationships, and adaptation or flexibility, which is likely because those were the most developed during the pandemic. The findings provide useful insights for human resources management in the hospitality sector. The study points to good practices aiming to address the real development needs of hospitality workers.

**Keywords:** hospitality sector; soft skills; hard skills; human resources; workers' perceptions; COVID-19 pandemic; comparative study

## 1. Introduction

The tourism sector was hit particularly hard by the impacts of the COVID-19 pandemic, suffering a sharp decline in 2020 and 2021. However, the international flow of tourists around the world in 2022 shows that despite its great vulnerability, the industry has a strong potential for recovery after periods of crisis. On the other hand, this resumption has been accompanied by a severe lack of skilled workers in the hospitality sector, due to the massive transfer of its labour force to less severely affected activities by the pandemic crisis. In parallel, the hotel sector, especially the four- and five-star hotels, increasingly offer a personalised and experiential service before, during and after the stay. Therefore, it is of utmost importance that people working in the hotel sector have certain skills, such as communication and interpersonal skills, an ability to generate empathy, digital skills, flexibility, and the ability to work in a team and to manage conflicts. In this context, the present study aims to compare hotel workers' perceptions—namely, their confidence of and perceived need to improve each of the aforementioned skills—during lockdown (in 2020) and during the resumption of tourism activity in Portugal (in 2022).

To reach this goal, four- and five-star hotels located in the historic centre of Porto (Portugal) have been adopted as study settings. The area was considered an ideal habitat for this research due to its importance as a tourism product. Porto has been elected Europe's best city destination by the World Travel Awards (WTA 2022) several times. That includes the 2022 edition, and a year in which the Portuguese hospitality industry's performance is already surpassing that of 2019, which had been the greatest so far (Turismo de Portugal

2022). Recognized as a Cultural Heritage of Humanity by UNESCO since 1996, the HCP plays a significant role in this achievement.

The present study's relevance and originality stem from the combination of the addressed subject (soft and hard skills) and the context in which it is addressed (i.e., Portugal's state of emergency due to the COVID-19 pandemic and the following period of tourism resumption). Naturally, previous studies have approached the importance of intellectual capital in hotels (Astuti et al. 2020), and even in the hotel industry in the HCP (Magalhães et al. 2022). However, the recent crisis likely led to a change in the need for qualification. During the pandemic, new needs and perceived benefits arose, demanding specific competencies from hospitality workers. Naturally, some of those changes are structural, and persist after the main crisis. Therefore, new, up to date studies are necessary to understand the role of hard and soft skills in catering to guests' needs and the expected benefits.

## 2. Literature Review

### 2.1. The Hotel Sector in Porto and Northern Portugal

In 2019, tourism in Porto and Northern Portugal was growing above the national average, which in turn, was already higher than the EU average. The area was achieving record numbers of overnight stays (which had increased 10.6% in comparison to the previous year), revenue (which had increased 14.8%), and RevPAR (revenue generated per available room). In this context, ten new hotels were opened in Porto that year (adding 960 new rooms to its hospitality offer), mostly due to the additional demand generated by the MICE (Meetings, Incentives, Conferences and Exhibitions) segment. This growth increases competitivity in the hospitality industry, raising the demand for excellent services, and therefore, for qualified professionals. Hotels employed 22,472 people in Porto, in 2018 (a 33% increase in comparison to 2010). Most of this workforce, however, have quite a low level of formal qualification, as by 2019, only 13% had completed university studies, and more than half (53%) had only completed basic studies.

Naturally, the tourism sector was hit particularly hard by the pandemic. In Portugal, guest numbers decreased 61.3% in 2020, in comparison to 2019, and overnight stays decreased 63%, leading to an over 66% decline in the hotels' total income. Accordingly, the Portugal National Bank (2021) revealed that tourist revenues in 2020 were less than half (57.6% less) of those achieved in 2019. Naturally, this recession scenario had an impact on employment in the hospitality industry, which decreased 8.9% (INE 2021). The total number of people formally employed, however, does not tell the whole story, as according to the Portuguese Hotels Association (2021) (AHP), 90.27% of Portuguese hotels laid off part or all of their workers. Therefore, even amongst those still officially employed, most had their income severely reduced. Amongst those still working, the number of employees with only basic education increased 2% (reaching 55% of the total) (INE 2021). This is likely related to the closure of higher end hotels and restaurants, which normally need to hire more qualified professionals.

In 2022, when the confinement had been lifted in most counties, the tourism industry experienced an upturn, and Porto was no stranger to this phenomenon. The numbers of the tourism industry in 2022 are surpassing those of 2019. In the accommodation sector, 3.4 million guests and 9.9 million overnight stays were recorded in August 2022. Compared to August 2019, there were increases of 1.2% and 2.8%, respectively (INE 2021). This is the first time since the start of the pandemic that increases have been recorded compared with the corresponding period before the pandemic. However, despite this growth, the hotel sector is currently going through a labour crisis, as during the lockdown period, workers were forced to look for jobs in other industries. Higher education in Tourism was also affected by the pandemic, registering a general decrease in the demand for Tourism degrees.

In sum, on the one hand, the pandemic created new service standards and the expected values from consumers rose. On the other hand, it led to a human resources crisis, which is aggravated by a decrease in demand for formal education in the area. In this context, it is

important to understand the hard and soft skills required to keep hospitality businesses competitive in this post-COVID-19 era.

### 2.2. Hard and Soft Skills before and after Pandemics

The concept of skill is widely discussed in business management literature (Stevens 2012). In this context, a relevant distinction is that between soft and hard skills, which has been addressed by several authors (Koval et al. 2020; Sopa et al. 2020; Wibowo et al. 2020). Moreover, several studies (Hersey and Blanchard 1977; Katz 1974) point to the importance of a leader having both types of skills (soft and hard), which is reflected on the profile of professionals sought out by organizations (Müller and Turner 2010). In this context, Parry (1996) states that hard skills are the essential competencies for performing a particular task, that is, technical abilities. Soft skills, in turn, are related to emotional competences.

Consequently, hard skills have a mainly cognitive nature and imply the previous acquisition of some type of knowledge (North and Worth 2000; Weber et al. 2009; Windels et al. 2013), which should be applied in each circumstance (DeLong and Elbeck 2018). Soft skills, in turn, are interpersonal and behavioural traits, which help people put their hard skills into practice (Windels et al. 2013). Some authors argue that, with the pandemic, soft skills became the new hard skills (Trudeau-Poskas 2020), increasing not just in terms of valorisation by companies, but also in measurability (Devedzic et al. 2018).

In terms of development, hard skills can be acquired through professional experience, academic training, and lifelong training (Katz 1974; Leroux and Lafleur 1995; Maniscalco 2010; Rao 2012). Examples of hard skills are linguistic competencies, such as English (Suebwongsuwan and Nomnian 2020), and digital competencies (Kolobkova et al. 2021). In fact, the importance of digital skills has become even more evident with the pandemic, especially due to the context of remote work (Ferreira et al. 2022; Zancajo et al. 2022).

Soft skills are transversal, as they are related to personality, behaviour and knowing how to be (Heckman and Kautz 2012). They also help people realize their true potential, and thus, promote the achievement of goals (Muir 2004), being thus essential for leadership functions (Jamison 2010). Several authors also argue that soft skills are not so easily measurable (Maniscalco 2010; Raftopoulos et al. 2009; Rao 2012; Shakir 2009). Some of the most relevant soft skills in the labour market are communication, critical spirit, flexibility and adaptability, ability to work in a team and relate to others, ability to solve problems and negotiate, ability to make decisions, self-confidence, self-management, ethics (Ajzen 1991; Patacsil and Tablatin 2017; Pritchard 2013; Robles 2012; Singh and Singh 2008; Williams 2015), and autonomy (Windels et al. 2013). Additionally, the importance of several soft skills, namely the ability to work in a team, communication skills, an ability to solve problems, and critical spirit, increased with the pandemic (Michel et al. 2022). Regarding communication, the pandemic accelerated its shift to digital platforms (Nguyen et al. 2020). Other soft skills, such as leadership, adaptability and flexibility, empathy, relationship building, openness to new technologies, and openness to change, also gained importance during the pandemic (Pezer 2021).

In sum, if competences were already essential for the success of organizations before the pandemic (Moldoveanu and Narayandas 2019), their importance has become even more evident after this crisis (Ferreira et al. 2022; Ratten 2020). Indeed, recent studies show that the importance of reskilling and upskilling, particularly in digital contexts, has increased in the post-pandemic period (Verma and Gustafsson 2020). In addition, currently, the human resources of companies have greater and faster access to tools that allow them to obtain new skills and retrain others (Jaiswal et al. 2021). At the same time, it is necessary to be aware that not everything is transferable to digital, and consequently, soft skills continue to be a differentiating factor (Black and van Esch 2020).

Most of these trends apply to the hospitality industry. However, due to its vulnerability to crises, as well as to particularities of tourism services—inseparability, intangibility, and high fixed operation costs (Middleton et al. 2009)—the impact of the pandemic was especially hard. This impact is addressed in more detail in the next section.

*2.3. Workers Self-Evaluation of Competencies (Comparison during and after)*

The investigation of skills in the hospitality sector has been addressed by several authors (Awasthi et al. 2020; Hahang et al. 2022; Hertzman et al. 2015; Kiryakova-Dineva et al. 2019; Magalhães et al. 2022; Mohammad 2020; Spowart 2011; Susaeta et al. 2020; Weber et al. 2013, 2020). However, due to the changes brought about by the pandemic, which severely affected this industry, it is urgent to reflect upon the skills needed in this new context (Hahang et al. 2022).

Regarding soft skills, recent studies point to the importance of leadership and teamwork (Hahang et al. 2022; Jawabreh et al. 2020). More specifically, Hahang et al. (2022) show that positive thinking, the ability to make decisions, flexibility, critical thinking, the ability to build trusting relationships, and the ability to communicate had a beneficial impact on leaders during the pandemic. Additionally, Fang et al. (2017) state that leadership must consider individual characteristics. In terms of flexibility, this was one of the skills that gained the most relevance with the pandemic, having become key for companies to adapt to the new reality and face critical circumstances (Barry et al. 2022).

Regarding hard skills, digital skills were already essential before the pandemic, and became even more important afterwards (António and Rita 2021; Ndou et al. 2022). On the other hand, studies also demonstrate that hospitality companies often have deficiencies in terms of these skills (Parsons et al. 2022). In this context, Carlisle et al. (2021) draw attention to the gap between existing and required digital skills.

The coordination between soft and hard skills is particularly relevant in the tourism sector, namely in a scenario of digital transformation and great instability (Strietska-Ilina and Tessaring 2005). In the past, organisations valued professionals' technical skills almost exclusively. Currently, this is no longer the case, especially in the hospitality industry. Therefore, the search for soft skills increased, particularly in the case of the ability to communicate and work in a team, flexibility, the ability to manage time, emotional intelligence and a sense of empathy, an ability to solve problems, positive attitude (Sonnenschein 2021), and conflict solution skills (Erdly and Kesterson-Townes 2003; Kay and Russette 2000). In fact, studies conducted before the pandemic described the ability to communicate, work in a team and leadership as the most important skills for the hospitality sector. More recent studies, which aimed to determine the most sought-after skills in the post-pandemic hotel industry, indicate that communication skills and the ability to work in a team and to solve problems are the most required skills in this sector (Scarinci et al. 2022).

Studies also indicate that the hotel industry is concerned with hiring people who combine their social skills with techniques (hard skills), as both are critical in delivering excellent services and enabling quality experiences (Marneros et al. 2020). On the other hand, some studies suggest that hotel industry employees are not prepared to deal with critical situations, requiring more training to develop certain social skills (Chalupa and Chadt 2021). Therefore, for understanding whether tourism sector workers have the demanded soft and hard skills, it is essential to provide competitive services in this post-crisis period, so that deficiencies and improvement opportunities can be identified and addressed.

## 3. Materials and Methods

This investigation's main objective was to compare hospitality workers' confidence of and the perceived need to improve their soft and hard skills during (2020) and after (2022) the lockdown imposed as a safety measure due the COVID-19 pandemic. To pursue this objective, a survey was carried out with workers of four- and five-star hotels in the Historical Centre of Porto (HCP). The selection of this area as study setting is justified by several reasons. Recognised as a Cultural Heritage of Humanity by UNESCO since 1996, the HCP is one of the most visited places in Portugal. In fact, the city of Porto has been elected Europe's best city destination by the World Travel Awards (WTA 2022) several times, including in the 2022 edition.

Like in other major destinations in Portugal, tourism in Porto and northern Portugal was thriving in 2019, presenting record numbers of both occupancy, RevPAR, and overnight

stays. In sum, 2019 had been the best year so far for tourism in Portugal, and 2020 was expected to be even better. For 2022, the data so far leads to expectations that tourism gains will be superior to 2019. In fact, monthly numbers show increases of over 30% in comparison with the same periods of 2019 (Turismo de Portugal 2022). In the HCP, specifically, the total number of hotels increased 68% between 2009 and 2019, according to INE. Therefore, the area has been playing a significant role in this recovery trajectory. Conceding this scenario, the HCP was deemed an adequate setting for the present study.

### 3.1. Participants

A survey questionnaire was used to collect data from four- and five-star hotel workers in the HCP. Therefore, the total research population consisted of workers of only 19 hotels, the total number of hotels within these categories in the HCP. As the first cross-sectional data collection was carried out in April and May 2020, that is, during the lockdown, all the procedures took place remotely. In this context, the first step consisted of contacting human resources directors via e-mail. The message contained a link to the questionnaire, which they were solicited to forward to workers under their supervision. For the second dataset, for the sake of consistency, a similar procedure was adopted.

The questionnaire employed both during and after the lockdown period included a description that aimed to ensure respondents that the information they provided was anonymous and confidential. This aimed to increase the response rate, as hotel managers are often insecure about sharing information they might view as critical.

Prior to the first data collection phase, aiming to detect potential issues, a pre-test was carried out with a smaller sample of workers. The total sample included 101 valid responses, 48 within the during-lockdown section and 53 within the post-lockdown section. This relatively small sample is a result of the context of the study—namely, the collection of data during the lockdown period, when hotels were closed or with minimum activity. On the other hand, the limitations it could bring about are mitigated by the relatively small total research population (workers from 19 hotels). Although small, the sample includes workers from all different functions and levels, as detailed in Table 1, which summarises the sample characterisation.

### 3.2. Instrument

The questionnaires applied during and after the lockdown feature the same items of confidence of and perceived need to improve soft and hard skills, to allow for the intended comparisons. The evaluated skills and the wording of the items follow the rationale adopted by Magalhães et al. (2022). In sum, eight questions measured workers' evaluation of their own skills (namely how much they thought their skill level was enough to satisfactorily carry out their job), and another eight measured their perceived need to improve these same skills. The evaluated skills included five soft skills, three hard skills, and "other skills". The items were operationalised on a 5-point Lickert scale (1 = Strongly disagree; 5 = Strongly agree). The questionnaire ended with categorical questions on sociodemographic and job-related variables, which allowed for the sample characterization.

Data were collected in two distinct periods separated by two years: 2020 and 2022. Regarding analysis methods; first, the data were subjected to basic descriptive statistics for the sample characterization and general description of workers' evaluation on their skills. Then, reliability tests were performed on the sets of soft and hard skills, as they were later treated as dimensions for the sake of comparison. Finally, independent sample Mann–Whitney U tests (as the variables were not normally distributed) were employed to compare workers' confidence of and the perceived need to improve their soft and hard skills during and after the lockdown period.

**Table 1.** Sample comparison—during and post lockdown.

|  | During Lockdown | After Lockdown |
|---|---|---|
| **N** | 48 | 53 |
| **Gender** | | |
| Male | 52.5% | 54.7% |
| Female | 47.5% | 45.3% |
| **Formal education** | | |
| Higher education | 45.5% | 54.7% |
| Technical Superior Professional Course | 16.8% | 17.0% |
| Secondary school | 14.9% | 1.9% |
| Master's degree | 14.9% | 20.08 |
| Level 5 course | 4.0% | 1.9% |
| Basic school | 3.0% | 1.9% |
| PhD | 1.0% | 1.9% |
| **Age** | | |
| 29 to 39 | 41.6% | 41.5% |
| 18 to 28 | 33.7% | 28.3% |
| 40 to 50 | 16.8% | 20.8% |
| 51 to 61 | 6.9% | 7.5% |
| 62+ | 1.0% | 1.9% |
| **Department** | | |
| Reception | 25.0% | 34.0% |
| Food and beverages | 18.8% | 13.2% |
| Housekeeping | 16.7% | 0.0% |
| Maintenance | 12.5% | 0.0% |
| Administration | 10.4% | 26.4 |
| HR | 8.3% | 11.3% |
| Marketing | 6.3% | 13.2% |
| Management | 2.1% | 1.9% |
| **Function or position** | | |
| Undifferentiated worker | 50% | 18.9% |
| Intermediate management | 14.6% | 32.1% |
| Technical | 10.4% | 11.3% |
| Administrative | 8.3% | 11.3% |
| Auxiliary | 8.3% | 1.9% |
| Top management | 6.3% | 20.8% |
| Intern | 2.1% | 3.8% |
| **Type of contract** | | |
| Fixed term contract | 43.8% | 43.4% |
| Uncertain term contract | 18.8% | 3.8% |
| Unlimited term contract | 18.8% | 43.4% |
| Partial contract | 14.6% | 1.9% |
| Internship | 2.1% | 3.8% |
| Outsourced | 2.1% | 3.8% |
| **Time in the company** | | |
| 1 to 3 years | 43.8% | 26.4% |
| Less than 1 year | 22.9% | 22.6% |
| 3 to 5 years | 18.8% | 18.9% |
| More than 5 years | 14.6% | 32.1% |

## 4. Results

### 4.1. Sample Characterization

To allow for a comparative study of workers' confidence and perceived need to improve their soft and hard skills during and after the lockdown period, two different samples were collected in April 2020 and April 2022, respectively. Table 1 summarizes the sociodemographic profile and work situation of each sample.

*4.2. Workers Self-Evaluation of Competencies (Comparison during and after)*

Both during and after the lockdown, most workers feel that they have the necessary skills to satisfactorily carry out their jobs. During the lockdown period, the skill with which workers showed the most confidence was teamwork (4.58), followed by interpersonal relationship competencies (4.46) and adaptation or flexibility (4.33). At the bottom end, workers showed they were least confident about their digital competencies (4.00), followed by other competencies (4.19) and specific competencies of the work area (4.23). After the lockdown period, the competence with which workers are the most confident is once again teamwork (4.81). However, this time it is followed by adaptation and flexibility (4.70) and communication (4.66), which had come in fifth in the during lockdown sample. Interpersonal relationship competencies, second place within the lockdown sample, now came in fourth (4.64). The competencies with which workers are the least confident are the same as during the lockdown period: conflict management (4.25), others (4.3) and specific competencies of the work area (4.40). Both during and after the lockdown period, the three competencies with which workers were the most confident were all soft competencies. Meanwhile, the three competencies with which they were the least confident were all hard skills.

Regarding differences between the two periods, as shown on Table 2, after the lockdown, workers evaluated their digital, communication, and teamwork skills significantly better. That is, workers are particularly more confident of their soft skills, as those include two (communication, teamwork) out of the three that showed significant improvement. Regarding the other competence categories, averages are consistently higher in the after-lockdown sample, but with no statistical significance.

**Table 2.** Differences in the self-evaluation of competences—during and after lockdown.

|  |  | Mean | SD | Mean Difference | *p* |
|---|---|---|---|---|---|
| Linguistic competencies | During lockdown | 4.33 | 0.808 | −0.176 | 0.226 |
|  | After lockdown | 4.51 | 0.750 |  |  |
| Digital competencies | During lockdown | 4.00 | 1.011 | −0.509 | 0.005 * |
|  | After lockdown | 4.51 | 0.639 |  |  |
| Communicational competencies | During lockdown | 4.29 | 0.771 | −0.369 | 0.009 * |
|  | After lockdown | 4.66 | 0.517 |  |  |
| Interpersonal relationship competencies | During lockdown | 4.46 | 0.713 | −0.183 | 0.272 |
|  | After lockdown | 4.64 | 0.484 |  |  |
| Conflict management competencies | During lockdown | 4.25 | 0.700 | 0.005 | 0.890 |
|  | After lockdown | 4.25 | 0.757 |  |  |
| Teamwork competencies | During lockdown | 4.58 | 0.613 | −0.228 | 0.046 * |
|  | After lockdown | 4.81 | 0.395 |  |  |
| Flexibility and adaptability competencies | During lockdown | 4.44 | 0.741 | −0.261 | 0.087 |
|  | After lockdown | 4.70 | 0.463 |  |  |
| Specific competencies of the work area | During lockdown | 4.23 | 0.751 | −0.167 | 0.166 |
|  | After lockdown | 4.40 | 0.793 |  |  |
| Other competencies | During lockdown | 4.19 | 0.532 | −0.114 | 0.250 |
|  | After lockdown | 4.30 | 0.668 |  |  |

* Significant at the 0.05 level.

*4.3. Perceived Need to Improve Competencies during and after Lockdown*

Regarding workers' perceived need to improve their skills, values are closer to the centre of the scale. During the lockdown period, linguistic competencies (3.38) were those the workers felt they needed to improve the most, followed by digital competencies (3.35) and other competencies (3.31). In accordance with their self-evaluation, teamwork (2.67) was the competency workers least felt that they needed to improve, followed by adaptation and flexibility (2.73) and interpersonal relationship competencies (3.02). After the lockdown, the three competencies that workers felt they needed to improve the most were the same. However, digital competencies (3.79) and linguistic competencies (3.75) switched positions; that is, the former became the workers' priority in terms of skill development. Analogous to workers' evaluation of their own skills, in the post-lockdown period, the skills they felt they needed to improve the least remained the same as before: teamwork (3.34), adaptation and flexibility (3.36) and communicational competencies (3.40). Once again, in accordance with workers' evaluation of their skills, both during and after the lockdown period the three competencies that workers felt they must improve the most included two hard competencies (linguistic and digital) and one generic category (other competencies).

Accordingly, the three competencies that workers felt they need to improve the least are all soft skills. Regarding differences in the perceived need to improve competencies between the during and after lockdown samples, as shown in Table 3, workers' perceived a significantly higher need to improve their interpersonal relationship, conflict management, teamwork, and flexibility or adaptability skills, as well as those specific to the work area, and other competencies. That is, although generally, the workers are more confident and perceived a lower level of need to improve their soft skills, in the after-lockdown period, they are particularly more aware of the need to improve them as well. This is evidenced by the predominance of soft skills within those with a significantly higher level of perceived need to improve within the post-lockdown sample. Regarding the remaining categories, the average is consistently higher in the after-lockdown sample, but with no statistical significance.

**Table 3.** Perceived need to improve competencies—comparison: during and after lockdown.

|  |  | Mean | SD | Mean Difference | *p* |
|---|---|---|---|---|---|
| Linguistic competencies | During lockdown | 3.38 | 1.438 | −0.380 | 0.226 |
|  | After lockdown | 3.75 | 1.239 |  |  |
| Digital competencies | During lockdown | 3.35 | 1.263 | −0.438 | 0.081 |
|  | After lockdown | 3.79 | 1.063 |  |  |
| Communicational competencies | During lockdown | 3.10 | 1.242 | −0.292 | 0.211 |
|  | After lockdown | 3.40 | 1.230 |  |  |
| Interpersonal relationship competencies | During lockdown | 3.02 | 1.263 | −0.489 | 0.044 * |
|  | After lockdown | 3.51 | 1.219 |  |  |
| Conflict management competencies | During lockdown | 3.13 | 1.315 | −0.573 | 0.028 * |
|  | After lockdown | 3.70 | 0.972 |  |  |
| Teamwork competencies | During lockdown | 2.67 | 1.389 | −0.673 | 0.012 * |
|  | After lockdown | 3.34 | 1.239 |  |  |
| Flexibility and adaptability competencies | During lockdown | 2.73 | 1.180 | −0.629 | 0.006 * |
|  | After lockdown | 3.36 | 1.210 |  |  |
| Specific competencies of the work area | During lockdown | 3.13 | 1.231 | −0.611 | 0.010 * |
|  | After lockdown | 3.74 | 1.146 |  |  |

**Table 3.** *Cont.*

| | | Mean | SD | Mean Difference | *p* |
|---|---|---|---|---|---|
| Other competencies | During lockdown | 3.31 | 1.188 | −0.442 | 0.040 * |
| | After lockdown | 3.75 | 1.090 | | |

* Significant at the 0.05 level.

### 4.4. Confidence of and the Need to Improve Soft and Hard Skills

To further analyse the improvement in workers' self-evaluation of their competencies, as well as their increased perceived need to further enhance them after the lockdown, comparisons were also made by grouping the skill categories within soft and hard skills. Prior to these comparisons, reliability tests were carried out for both factors, to ensure they could be grouped as reliable scales. Both Cronbach's alpha values are above 0.7 (0.747 for hard skills and 0.810 for soft skills), Nunnally's (1978) threshold for factor reliability, which suggests the scales are indeed reliable. As shown in Table 4, workers are significantly more confident of their hard skills after the lockdown. Soft skills also present a higher average, but with no statistical significance. Nevertheless, the workers also show a higher level of perceived need to improve both their hard and soft skills. In sum, workers are more confident of their skills after the lockdown, and possibly also more eager to enhance them.

**Table 4.** Comparison of self-evaluation and perceived need to improve soft and hard skills—during and after lockdown.

| | | Mean | SD | Mean Difference | *p* |
|---|---|---|---|---|---|
| **Confidence in soft skills** | During lockdown | 4.40 | 0.553 | −0.207 | 0.091 |
| | After lockdown | 4.61 | 0.377 | | |
| **Confidence in hard skills** | During lockdown | 4.19 | 0.701 | −0.284 | 0.028 * |
| | After lockdown | 4.47 | 0.594 | | |
| **Need to improve soft skills** | During lockdown | 2.93 | 1.146 | −0.531 | 0.012 * |
| | After lockdown | 3.46 | 1.051 | | |
| **Need to improve hard skills** | During lockdown | 3.28 | 1.161 | −0.476 | 0.038 * |
| | After lockdown | 3.76 | 0.948 | | |

* Significant at the 0.05 level.

## 5. Discussion

The goal of this study was to compare hospitality workers' confidence of and perceived need to reinforce their soft and hard skills during and after the lockdown period. To this end, a questionnaire survey was conducted with four- and five-star hotel workers in the Historical Centre of Porto both during (2020) and after (2022) the lockdown period. The questionnaires operationalised workers' confidence of and the perceived need to improve their soft and hard skills through items proposed based on previous contributions on the topic.

Previous studies showed that the most required soft skills before the pandemic were: ethics, ability to communicate and relate to others, critical spirit, ability to make decisions, ability to negotiate and solve problems, self-confidence, self-management (Ajzen 1991; Patacsil and Tablatin 2017; Pritchard 2013; Robles 2012; Singh and Singh 2008; Williams 2015), and autonomy (Windels et al. 2013). After the pandemic, studies point to the importance of teamwork, communication, problem-solving, leadership, interpersonal relationships, adaptability, flexibility, critical thinking (Michel et al. 2022), empathy, openness to new technologies, and openness to change (Pezer 2021).

In the hotel industry, studies point to teamwork and leadership skills as the core competencies (Hahang et al. 2022; Jawabreh et al. 2020). Studies addressing the post-pandemic period, point to a prevalence of soft skills, since positive thinking, the ability to make decisions, flexibility, critical thinking, the ability to build trusting relationships, the ability to communicate (Hahang et al. 2022), positive attitude, empathy and emotional intelligence, time management, problem solving (Sonnenschein 2021), and conflict management skills (Erdly and Kesterson-Townes 2003; Kay and Russette 2000) have also become critical for the industry.

In the context of the present investigation, during the lockdown period, the skills with which workers showed to be the most confident were teamwork, interpersonal relationship competencies, and adaptation and flexibility, in this order. This is a good sign, since, as concluded by Hahang et al. (2022) and Jawabreh et al. (2020), these are some of the most important competencies for the sector. After the lockdown period, the competence with which workers are the most confident is, once again, teamwork. However, this time it was followed by adaptation and flexibility (rather than interpersonal relationships), and communicational skills. This may be related to the context of remote labour relations that arouse during the pandemic. The COVID-19 crisis led to the interruption and reduction in hotel activities, forcing their personnel to adapt to new ways of working in a context of market instability (Barry et al. 2022), in which new forms of communication (Dias et al. 2021), mainly remote (Nguyen et al. 2020), had to be used. The competence with which workers continue to be the least confident is conflict management, which is worrying, since it is a core competence for the hotel industry (Erdly and Kesterson-Townes 2003; Kay and Russette 2000).

Regarding the differences between confidence of skills before and after lockdown, workers showed they were significantly more confident of their teamwork skills. On the other hand, they also perceive a higher need to improve them, as they do with all other soft skills except for communication. This is likely due to the realisation that, as pointed out by Barry et al. (2022), the importance of these skills, particularly flexibility, has also increased because of the pandemic. Regarding hard skills, the workers showed they were least confident with their digital competencies, followed by other competencies and specific competencies of the work area. This finding is also a cause for concern in the hotel industry since digital skills are part of the most currently demanded hard skills by the market (António and Rita 2021). Moreover, studies point to a gap between the desired and the actual level of these skills in this sector (Parsons et al. 2022). In fact, the competencies with which workers were the least confident before the lockdown were hard skills (linguistic competencies and digital competencies).

In the after-lockdown period, workers evaluated their digital skills significantly better. Nevertheless, although these hard skills continue to be those in which workers perceive the greatest need to improve, digital skills now take a more prominent place, becoming workers' priority in terms of improvement. This is in line with recent study (i.e., António and Rita 2021; Ndou et al. 2022) findings, which show that these are the most required skills in the hotel sector, as well as those in which greatest deficiencies are observed (Carlisle et al. 2021; Parsons et al. 2022). In the same vein, as the perceived need to improve all hard skills was already high, they did not present any significant change after the lockdown.

## 6. Conclusions

Considering the addressed findings, this study contributes to the theory of the field by reinforcing previous conclusions. In sum, the findings show that workers are more confident of their skills, but also more aware of the need to develop them, which is true for both soft and hard skills. However, the competencies that workers feel the least need to improve are soft skills, which is likely because they were the most developed during the pandemic. Therefore, the study corroborates previous contributions that point to a shift in the valorisation of workers' skills in the hospitality sector, namely, an increasing prevalence of soft skills, especially after periods of crisis.

In terms of practical contributions, the findings provide useful insights for human resources management in the hospitality sector. Namely, the study points to good practices aiming to address the real development needs of hospitality workers. For example, hotels should develop training plans that are more in line with the skills workers felt they needed to improve the most, particularly those they felt even more need to improve in the post-pandemic period (i.e., digital, and communicational competencies). Hotel human resources management should also consider those deficiencies in their recruiting programs. Other practices suggested by the present findings are performance evaluation systems that analyse the effectiveness of upskilling and reskilling, and the development of career management programs more suited to the reality of each employee.

With this awareness and action, companies in this sector should be able to improve the qualification of their workforce, contributing to the improvement of motivation levels, the attraction of people with an adequate profile and, consequently, greater market competitiveness. Still in practical terms, this study helps potential jobseekers in the hospitality sector to invest in their training in areas considered more urgent, such as digital skills.

Despite its contributions, the present study also presents shortcomings. The main limitation of this research is clearly the sample size. This is explained, however, by the context in which it took place, when hotels were severely restricted by the pandemic (specially the first data collection). In addition, despite having two distinct data collection sections separated by a two-year period, the study is still of a cross-sectional nature. Therefore, it is still unclear whether this "new reality" of workers' perceptions about their skills represents a structural change, or whether it can be generalised to other contexts.

To overcome those limitations, future studies should adopt similar methods and variables to address hotel workers' perceptions of their skills in the future. A new assessment in three years, for example, could help understand such perceptions in a period when companies and workers had enough time to invest in training in the most critical areas, such as digital skills. Accordingly, these variables should also be examined in other geographical contexts, namely in Portugal's central and southern regions, as well as in other countries.

**Author Contributions:** Conceptualization, C.M., A.A. and M.I.A.-M.; methodology, C.M., A.A. and M.I.A.-M.; validation, A.A.; formal analysis, C.M. and A.A.; investigation, C.M. and M.I.A.-M.; resources, C.M.; data curation, A.A.; writing—original draft preparation, C.M. and M.I.A.-M.; writing—review and editing, C.M., A.A. and M.I.A.-M.; visualization, C.M., A.A. and M.I.A.-M.; supervision, C.M., A.A. and M.I.A.-M.; project administration, C.M., A.A. and M.I.A.-M. All authors have read and agreed to the published version of the manuscript.

**Funding:** This research received no external funding.

**Informed Consent Statement:** Informed consent was obtained from all subjects involved in the study.

**Data Availability Statement:** Data supporting the reported results (the dataset elaborated from the responses to the survey) can be directly requested to the corresponding author.

**Conflicts of Interest:** The authors declare no conflict of interest.

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
