# Peer review of "How Do Hospitality Workers Perceive Their Work Skills before and after the Lockdown Imposed by the COVID-19 Pandemic?"

_socsci, doi:10.3390/socsci11120588_

Round 1

Reviewer 1 Report

The paper is theoretical and empirical in nature. It has been prepared carefully and contains the elements necessary for a scientific article. The layout of the article is correct and traditional, and consists of a theoretical, methodological and empirical part.

The introduction introduces the subject and emphasizes the contribution of paper to science.

The paper contains a proper literature review, taking into account international research from recent years. The sources were selected correctly.

The methodology defines the method of conducting the research and characterizes the research group. However, this part is vague and should be completed:

- it was not explained why these statistical analysis tools were used. Although the simplest tools were used - mainly descriptive statistics and t-test, this was not specified in Mathod section. According to t-test analysis, but it can be applied when the test statistic would follow a normal distribution. Has a distribution study been carried out? If the responses do not form a normal distribution, other non-parametric tests, such as the Mann-Whitney test, had to be used.

- The authors do not indicate reliability analysis for hard and soft skills. Even the most popular study - Cronbach's alpha test - has not been shown as a measure of internal consistency

- there is only main aim in this paper - no research questions or hypotheses - it seems to be a consequence of too simple statistical methods used for the analysis, but it seems not very attractive to the potential reader. 

-The authors should consider more advanced methods of statistical analysis. The current level of analysis is rather poor.

The authors indicate mainly a small research group as research limitations, while it seems that the research process is a much bigger problem. Authors should define more clearly what prevents the analysis of the results of quantitative research from providing more advanced information, e.g. on correlation or including them in the regression model. It should be remembered that the limited analysis also means general and not quantitatively verified conclusions. I suggest developing the analytical part.

Author Response

Please, find a point-by-point response to the reviewer’s comments in the attached file.

Reviewer 2 Report

Dear Author(s),

The abstract needs to be explained more about the results. The discussion section is very well written and explained but is not reflected in the abstract. 

I mean you mentioned how the soft skill types were different before and after the pandemic and you need to mention that in the abstract to present the value of your research. Otherwise, it is obvious that there must be some differences!

The discussion part is clear and well-written but it is better to be separated by the conclusion. Also, you mentioned the implications and limitations in 'Discussion and Conclusion', then you need to rearrange the final sections. 

Author Response

(The authors gave the same response as above.)
